# LABEL SPACE-INDUCED PSEUDO LABEL REFINEMENT FOR MULTI-SOURCE BLACK-BOX DOMAIN ADAPTATION

## ABSTRACT

Unsupervised Domain Adaptation (UDA) aims to train a model for an unlabeled target domain by transferring knowledge from a source domain. However, standard UDA requires access to source data and models, prohibiting its practical application in terms of privacy and security. Black-Box DA (BDA) reduces such constraints by defining a pseudo label from a single source prediction, which allows for self-training of the target model. Nonetheless, existing methods have limited consideration for multi-source settings, in which multiple source domains are available to generate pseudo labels. In this work, we introduce a novel training framework for multi-source BDA (MSBDA), dubbed Label Space-Induced Pseudo Label Refinement (LPR). Specifically, LPR incorporates a Pseudo label Refinery Network (PRN) that learns the relation between each source conditioned by the target from source predictions. The target model is adapted by self-learning using a pseudo label generated by PRN. We provide theoretical supports for the performance of the LPR. Experimental results on four benchmark datasets demonstrate that MSBDA using LPR achieves highly competitive performance compared to state-of-the-art approaches with different DA settings.

## 1 INTRODUCTION

Unsupervised domain adaptation (UDA) is used to transfer domain knowledge acquired from a labeled source domain to an unlabeled target domain. The primary goal of UDA is to mitigate the impact of distribution shifts between the source and target domains, while reducing the labeling burden in the target domain (Saito et al., 2018; Long et al., 2015; Sun & Saenko, 2016; Long et al., 2017). However, existing UDA methods still demand labeled source data (Ganin et al., 2016; Long et al., 2015) or the parameters of a source model (Qiu et al., 2021; Liang et al., 2020) to train a target model. These requirements impede the deployment of a target model in real-world applications, and they also raise concerns about privacy and security (Dong et al., 2020; Jaradat, 2017).

Black-box DA (BDA) (Zhang et al., 2021; Liang et al., 2022; Liu et al., 2022b;a; Yang et al., 2022) has been proposed for a target model to adapt using only the prediction of a source model in an unlabeled target domain. No other knowledge from the source domain is utilized. Due to the limited resources, a straightforward BDA approach would generate pseudo labels from a source model and directly use them for adaptation. The primary concern is the qualities of pseudo labels. Liu et al. (2022b;a) created a pseudo label through a weighted sum of source and target prediction, in which the weight parameters changed with the confidence of a target prediction. Liang et al. (2022) tried to reduce noise in source prediction using adaptive label smoothing.

Previous BDA methods have used a single source model, assuming that the distributions of source and target domains would be sufficiently correlated. It is natural and necessary to consider multiple source domains in BDA (MSBDA), because a user can select one or more source APIs. However, the correlation of source domains is usually unknown for the target adaptation, which could be more realistic but challenging than the BDA. There are only few MSBDA studies Liang et al. (2022). Liang et al. calculated the average of the individual source predictions to extend their original BDA method to an MSBDA method. However, they ignored the different importance of source models.

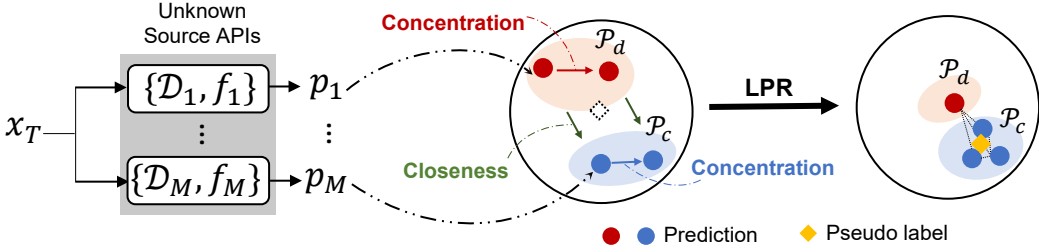

Figure 1: Illustration of Multi-Source Black-box DA setting (MSBDA) and the proposed self-learning framework to concentrate source prediction within a label space and make different label spaces close to refine a pseudo label, based on a theoretical analysis.

In this paper, we propose a novel self-supervised pseudo-labeling framework for MSBDA to effectively train a target model. While existing BDA methods have neglected to consider various characteristics of different sources, the proposed method focuses on exploring statistical relations with the source domains and extracting useful information from them. According to our theoretical analysis, a risk factor of a source prediction, which expresses how a target error of a hypothesis is deviated from an oracle error, can determine a positive or negative impact on the efficacy of pseudo label generation. Based on this, we propose a pseudo-label refinery network (PRN) with the division of a label space to produce confident source predictions to facilitate positive knowledge transfer. The optimization is conducted by focusing on the concentration of source predictions within a label space and closeness to other label spaces as presented in Figure 1, motivated by our theoretical analysis. In contrast, assigning an equal contribution on every prediction would be sub-optimal, because some predictions may have negative impacts.

The main contributions of this work include as follows:

- We develop a novel MSBDA framework that leverages only the predictions of source models to explore positive knowledge from multiple source domains. To the best of our knowledge, this work is the first to explore positive knowledge from multiple source domains, which is a challenging yet significant task in the field of MSBDA.

- We present a theoretical analysis to demonstrate the effectiveness of the proposed training strategy and propose the PRN architecture, specifically designed to resolve complex relations in source and target domains and refine a pseudo label.

- We evaluate the proposed method on four benchmark datasets and demonstrate that it outperforms state-of-the-art methods in various domain adaptation settings.

## 2 RELATED WORKS

**UDA.** Conventional UDA aims to adapt a target model in an unlabeled target domain, by leveraging the learned knowledge from a labeled source domain. Several studies (Long et al., 2015; Sun & Saenko, 2016; Long et al., 2017; Yan et al., 2017; Saito et al., 2018; Lee et al., 2019) have attempted to minimize the statistical discrepancy between source and target domains. Meanwhile, adversarial learning-based UDA methods have been presented to align source and target domains in feature-level (Tzeng et al., 2017; Long et al., 2018), pixel-level (Bousmalis et al., 2017; Sankaranarayanan et al., 2018; Xu et al., 2020b), and category-level (Saito et al., 2018; Xie et al., 2018; Pan et al., 2019; Xu et al., 2020a). These methods demand to access source data or the parameters of a source model.

**Multi-source domain adaptation (MSDA).** MSDA is an extension of the standard DA, when it is unclear which source domains are best suited for a target adaptation. In (Mansour et al., 2008), the distribution of a target domain was approximated through a mixture of those of source domains. (Hoffman et al., 2018a; Zhao et al., 2018; Li et al., 2018) derived theoretical cross-domain bounds to model the discrepancy among multiple source domains. (Zhao et al., 2018) proposed multiple domain adversarial networks (MDAN) to learn invariant features to various sources. (Xu et al., 2018)

presented a deep cocktail network (DCTN) to address category shifts. (Peng et al., 2019) developed a dynamic method to align the moments of source and target feature distributions.

**Source-free domain adaptation (SFDA).** SFDA further eliminated the accessibility to raw source data from the UDA setting and used pseudo-labeling as an enabling method (Liang et al., 2020; Qiu et al., 2021; Ahmed et al., 2021). (Qiu et al., 2021) used confidence re-weighting and regularization to reduce the negative transfer by noisy pseudo labels. (Ahmed et al., 2021) introduced a solution for multi-source SFDA, by combining the source models with suitable weights. Their model achieved comparable performance to the best choice of a single source model.

**BDA.** There have been several studies (Zhang et al., 2021; Liang et al., 2022; Liu et al., 2022b;a; Yang et al., 2022) to solve the BDA problems. In Liang et al. (2022), an adaptive label smoothing and a structural knowledge distillation have been proposed to approximate a target prediction to source predictions. In Liu et al. (2022b;a), a level of confidence has been calculated for pseudo-labeling and a target adaptation. Our work is substantially different from the previous studies, when bridging the BDA and multi-source setting.

**Pseudo-labeling.** It has been widely used for UDA to overcome the lack of labeled data in a target domain, when the source data or model are not directly accessible. Self-training is applied to produce a pseudo label using a source prediction and use it to fine-tune a target model. (Liang et al., 2022) used pseudo-labeling in a black-box setting. Since these methods assumed single-source DA, they were not directly applicable to MSDA. Xu et al. (Xu et al., 2018) and Wang et al. (Wang et al., 2020) used hard pseudo-labeling to learn the interaction among domains. Different from these studies, we use a deep model to apply weighted distribution through self-attention and refining pseudo-labels.

## 3 METHODOLOGY

### 3.1 PROBLEM FORMULATION

We address an adaptation of a $K$-way classification model in an MSBDA setting. There are $M$ labeled source domains $\mathcal{D}_S = \{\mathcal{D}_1, \mathcal{D}_2, \ldots, \mathcal{D}_M\}$ and one unlabeled target domain $\mathcal{D}_T$. $\mathcal{X}_i$ and $\mathcal{Y}_i$ refer to the set of input samples and their annotations from $\mathcal{D}_i$, respectively. We assume that the source domain and target domain share the same label space, *i.e.*, $\mathcal{Y}_i = \mathcal{Y}_T$ for all $i$. In contrast, $x_i \in \mathcal{X}_i$ and $x_T \in \mathcal{X}_T$ display different distributions. A source model $f_i \in f_S = \{f_1, f_2, \ldots, f_M\}$ has been trained using $\mathcal{D}_i$ and used in a target domain. Then, $p_i = f_i(x_T) \in \mathcal{P}$ is the only means to generate a pseudo label $\hat{p}$ for the target adaptation.

The previous methods (Xu et al., 2018; Wang et al., 2020) relied on a hard decision using $p_i$ to generate a pseudo label. The proposed method aims to reduce an upper bound of a risk, which estimates how a target error of a hypothesis is deviated from the oracle error, to produce an appropriate pseudo label. We explain the theoretical foundation and the proposed adaptation mechanism in the following sections. All the proofs of theorems are provided in the supplementary material.

### 3.2 THEORETICAL ANALYSIS

Denote $y_T$ as the ground-truth label of a target sample, which is unknown, and $h \in \mathcal{H}$ as a hypothesis of a target model, respectively. Given a pseudo label $\hat{p}$, our goal is to find a theoretic upper bound of a difference between a target error $\epsilon(h, \hat{p}) = \mathbb{E}_{x \in \mathcal{X}_t}[|h(x) - \hat{p}|]$ and the oracle error $\epsilon(h, y_T) = \mathbb{E}_{x \in \mathcal{X}_t}[|h(x) - y_T|]$, because the minimization of the bound can serve as a risk mitigation strategy to ensure that $\hat{p}$ is reliable for adaptation. For $\hat{p}$, a weighted linear combination of each source output has been a popular choice (Hoffman et al., 2018a; Zhao et al., 2018; Li et al., 2018). Following the assumption, we derive a general upper bound (Yang et al., 2022) for the MSBDA as in **Theorem 1**.

**Theorem 1. (General upper bound of a risk in target prediction)** *Denote $h$ as a hypothesis in $\mathcal{H}$. We then establish a theoretical upper bound on the difference between the target error and the oracle error as*

$$|\epsilon(h, \hat{p}) - \epsilon(h, y_T)| \leq \sum_i \alpha_i \epsilon(p_i, y_T), \tag{1}$$

*where a pseudo label is defined as $\hat{p} = \sum_i \alpha_i p_i$, $\alpha_i \geq 0$, $\sum_i \alpha_i = 1$.*

Existing BDA methods (Liang et al., 2022; Liu et al., 2022a;b) have attempted to decrease empirical errors *e.g.* through de-noising of source predictions and label smoothing, when $y_T$ is not accessible. Instead, in this framework, we modify the general bound using tractable terms without the ground truth to provide a practical solution. For this purpose, we first define a representative prediction, denoted as $p_c$, which is assumed to be the one closest to the ground truth among the available source predictions. We then modify the theoretic upper bound, by considering $p_c$.

**Lemma 1. (Modified upper bound of a risk)**

$$|\epsilon(h, \hat{p}) - \epsilon(h, y_T)| \le \epsilon(p_c, y_T) + \eta, \tag{2}$$

*where $p_c = \arg\min_{p_i \in \mathcal{P}} \epsilon(p_i, y_T)$, and $\eta = \sum_i \alpha_i \epsilon(p_c, p_i)$.*

In the bound derived in **Lemma 1**, $\eta$ represents a degree of the dispersion of source predictions from the center at $p_c$ due to the weighted error terms. $\eta$ can be directly estimated with the accessible terms of source predictions, and the minimization of $\eta$ could reduce the upper bound of a risk associated with the reliability of a pseudo label as in Eq.(2). However, there would be some noisy outliers among source predictions to hinder the optimization. When most of $p_i$ are dispersed from $p_c$, $p_c$ needs to move away from $y_T$ to avoid the penalty, which leads to a sub-optimal solution.

To avoid the failure, we consider a division of a label space and decompose the bound into tractable terms. Let us assume there exist $\mathcal{P}_c$ and $\mathcal{P}_d \subset \mathcal{P}$ as two subsets of an entire label space. $\mathcal{P}_c$ and $\mathcal{P}_d$ are defined as the spaces, in which their samples are concentrated to $p_c$ and dispersed from $p_c$, respectively. They are mathematically defined as follows:

$$\mathcal{P}_c = \{p_i | \epsilon(p_c, p_i) \le \xi\}, \quad \mathcal{P}_d = \mathcal{P} \setminus \mathcal{P}_c, \tag{3}$$

where $\xi$ denotes a threshold for the label space division.

Then, we define the degree of the dispersion of each label space as below,

$$\eta_c = \sum_{p_i \in \mathcal{P}_c} \alpha_i \epsilon(p_c, p_i), \quad \eta_d = \sum_{p_i \in \mathcal{P}_d} \alpha_i \epsilon(p_d, p_i), \tag{4}$$

where $p_d = \arg\min_{p_i \in \mathcal{P}_d} \epsilon(p_c, p_i)$.

**Theorem 2. (Upper bound of a risk with a label space division)**

$$|\epsilon(h, \hat{p}) - \epsilon(h, y_T)| \le \epsilon(p_c, y_T) + \eta_c + \eta_d + \sum_{p_i \in \mathcal{P}_d} \alpha_i \epsilon(p_c, p_d), \tag{5}$$

*where $p_d = \arg\min_{p_i \in \mathcal{P}_d} \epsilon(p_c, p_i)$ is the representative prediction in $\mathcal{P}_d$. $\eta_c = \sum_{p_i \in \mathcal{P}_c} \alpha_i \epsilon(p_c, p_i)$ and $\eta_d = \sum_{p_i \in \mathcal{P}_d} \alpha_i \epsilon(p_d, p_i)$ are the dispersion of $\mathcal{P}_c$ and $\mathcal{P}_d$, respectively.*

As presented in Eq.(5), the reduction of both the dispersion in $\mathcal{P}_c$ and $\mathcal{P}_d$ and the distance between $p_c$ and $p_d$ would lower the upper bound. $\epsilon(p_c, y_T)$ is an inherent error in MSBDA. In the following sections, we will explain how to perform pseudo-labeling for a $K$-way classification task in a practical MSBDA setting, based on the theoretic analysis.

### 3.3 Label space division in $K$-way classification

In the $K$-way classification, given $x_T$, $p_{ij}$ refers to the $j$-th element of the output probability vector from the $i$-th source model $f_i(x_T) \in \mathbb{R}^K$ among $M$ source domains. We define a set $\mathcal{J} = \{j_{(i)}^*\}$ of indices $j_{(i)}^* = \arg\max_j p_{ij}$ to maximize $p_{ij}$ over $j$ and a set $\mathcal{I}_c = \{i_c | j_{(i)}^* = c\}$ of indices $i_c$ where $c = \arg\max_k \sum_i \mathbb{1}(j_{(i)}^* = k)$. We define a set $\mathcal{P}_c$ to include $p_{i_c j_c^*}$, in which $i_c \in \mathcal{I}_c$ and $j_c^* = j_{(i_c)}^*$. The representative label $p_c \in \mathcal{P}_c$ is selected as $\max p_{i_c j_c^*}$ over $i_c$. When the cardinal number of $\mathcal{I}_c$ is $M$, in which all the source predictions are different, we select the source prediction with the highest probability regardless of the categories.

$\mathcal{P}_d$ and $\mathcal{I}_d$ are defined as $\mathcal{P} \setminus \mathcal{P}_c$ and $\mathcal{I} \setminus \mathcal{I}_c$, respectively. $p_d \in \mathcal{P}_d$ is chosen as $\min p_{i_d j'}$ over $i_d$, when $j' \in \mathcal{J} \setminus \{j_c^*\}$ is the classification result of $f_{i_d}(x_T)$.

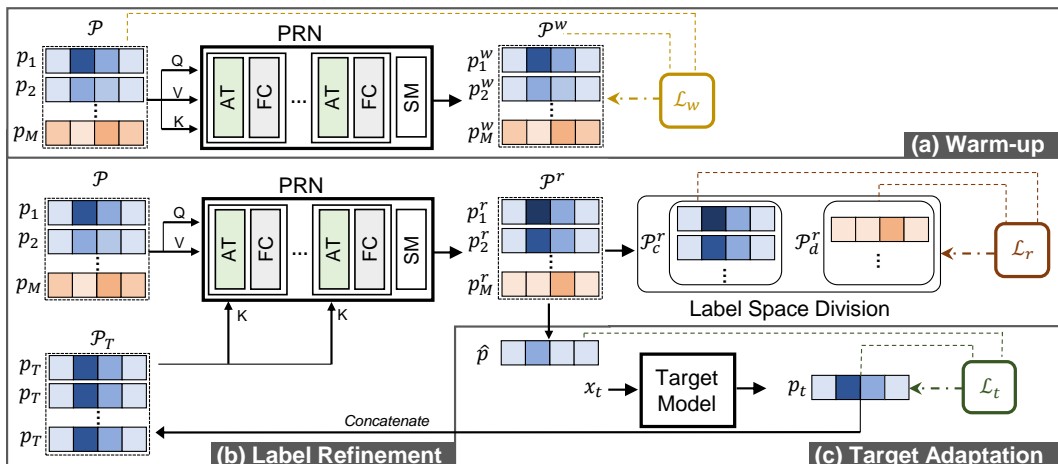

Figure 2: The proposed self-learning framework with a pseudo label refinement network (PRN), including a warm-up, a label refinement, and a target adaptation phase. PRN consists of attention (AT) and fully connected (FC) layers to consider the relevance between source and target domains and resolve their complex statistical relations. In the adaptation, the PRN is trained to improve the reliability of a pseudo label by encouraging the concentration within label spaces and closeness across label spaces, based on theoretical analysis.

## 3.4 Pseudo Label Refinery Network

The proposed method utilizes a pseudo label refinery network (PRN) with a target model to generate high-quality pseudo labels, as shown in Figure 2. PRN is designed to learn the relations not only between different source domains but also between source domains and a target domain. For this, it is implemented with the stacks of refinement blocks that include an attention layer (AT) and a fully connected layer (FC) followed by softmax (SM) layer, as presented in Figure 2. The AT is added to capture the level of attention or relevance of one source prediction to the other predictions. The FC generates refined predictions from the inputs.

When $p_i(= f_i(x_T))$ is digested to generate the final pseudo label through the PRN, it is not limited to be a simple linear combination of source predictions and $\alpha_i$. The refinement-and-adaptation process using PRN consists of three phases, including a warp-up phase, a label-refinement phase, and a target adaptation phase. It is highlighted that our genuine contribution in the PRN lies in both the warm-up and label-refinement phases. Although we have made modifications to the target adaptation phase compared to previous studies, we maintained consistent training parameters to assess the enhanced performance achieved by our proposed method.

### 3.4.1 Warm-up phase

The output of PRN is noisy with randomly initialized parameters in an early stage of adaptation. Warm-up phase is used to avoid a failure due to such noisy samples and to produce initial predictions similar to the original source predictions. Denote $\mathcal{P}$ and $\mathcal{P}^w$ as an input concatenation of source predictions $[p_1 \ldots p_M]^{\mathrm{T}}$ and the outcomes of the warm-up phase $[p_1^w \ldots p_M^w]^{\mathrm{T}}$, *i.e.*, $\mathcal{P}^w = \mathrm{PRN}(\mathcal{P})$. The PRN in the warm-up phase is trained using a loss function, defined as

$$\mathcal{L}_w = \mathbb{E}_{x_t \in \mathcal{X}_T} \mathbb{E}_{i \in \mathcal{I}} \mathcal{KL}(p_i \parallel p_i^w), \tag{6}$$

where $\mathcal{KL}$ is the Kullback-Leiger (KL) divergence, and $\mathcal{I}$ is a set of categorical indices. After the warm-up phase, $p_i^w$ is averaged over $i$ and used as an initial pseudo label to be refined later.

### 3.4.2 Label refinement phase

The PRN refines the input predictions using both source and target predictions in this phase. It is necessary to exploit both predictions, because a target prediction can offer knowledge learned from a

target model $f_T$ with an unlabeled target sample $x_T$. In what follows, the PRN takes the original source prediction $\mathcal{P}$ as a query and a value and a target prediction $p_T (= f_T(x_T))$ as a key and conducts a cross-attention operation through the AT and produces an output $\mathcal{P}^r$ as follows:

$$\mathcal{P}^r = [p_1^r \ldots p_M^r]^{\mathrm{T}} = \mathrm{PRN}(\mathcal{P}, \mathcal{P}_T), \tag{7}$$

where $\mathcal{P}_T = [p_T \ldots p_T]^{\mathrm{T}} \in \mathbb{R}^{M \times K}$. The cross-attention calculates the level of attention of one target prediction to several source predictions in the label space and allows the PRN to be trained in an unsupervised manner and reflect the relations between the source and target domains.

The label space $\mathcal{P}^r$ is divided into $\mathcal{P}_c^r$ and $\mathcal{P}_d^r$ in the label refinement phase in Figure 2. The majority of the pseudo labels that output the same classification results are grouped to $\mathcal{P}_c^r$. The other pseudo labels are grouped to $\mathcal{P}_d^r$. We then define a training objective based on the analysis in **Theorem 2**.

First, we define a concentration loss to reduce each dispersion within $\mathcal{P}_c^r$ and $\mathcal{P}_d^r$ (see $\eta_c$ and $\eta_d$ in Eq.(5)), respectively, given as,

$$\mathcal{L}_{cc} = \mathbb{E}_{x_T \in \mathcal{X}_T} \mathbb{E}_{p^r \in \mathcal{P}_c} \mathcal{KL}(p_c^r \parallel p^r), \tag{8}$$

and

$$\mathcal{L}_{cd} = \mathbb{E}_{x_T \in \mathcal{X}_T} \mathbb{E}_{p^r \in \mathcal{P}_d} \mathcal{KL}(p_d^r \parallel p^r), \tag{9}$$

where the representative labels $p_c^r \in \mathcal{P}_c^r$ and $p_d^r \in \mathcal{P}_d^r$ are chosen as explained in Section 3.3.

We employ a loss function to consider the distance between the two representative labels (see the last term in Eq.(5)), given as

$$\mathcal{L}_{ld} = \mathbb{E}_{x_T \in \mathcal{X}_T} \mathcal{KL}(p_c^r \| p_d^r). \tag{10}$$

Further, a stabilization loss $\mathcal{L}_s = \mathbb{E}_{x_t \in \mathcal{X}_T} \mathbb{E}_{i \in \mathcal{I}} \mathcal{KL}(p_i \parallel p_i^r)$ to produce an approximation of $p_i$ as an initial value and avoid overfitting of the PRN to irrelevant probabilities. The PRN is trained using the total loss $\mathcal{L}_r$ in the refinement phase, defined as

$$\mathcal{L}_r = \mathcal{L}_{cc} + \lambda_{cd}\mathcal{L}_{cd} + \lambda_{ld}\mathcal{L}_{ld} + \lambda_s\mathcal{L}_s. \tag{11}$$

### 3.4.3 Target adaptation phase

In this phase, a target model is trained using a generated pseudo label from the PRN. First, we compute an average of $p^r$ to decide the final pseudo label, *i.e.*, $\hat{p} = \frac{1}{M}\sum_{i=1}^{M} p_i^r$ and train a target model using $\hat{p}$ as as the ground truth through a self-learning loss, i.e., $\mathcal{L}_{sl} = \mathbb{E}_{x_T \in \mathcal{X}_T} \mathcal{KL}(\hat{p} \parallel f_T(x_T))$.

In addition, we utilize a mutual information loss to encourage the target model to maintain diversity among its predictions across all target instances. To this end, the mutual information objective (Liang et al., 2020; 2022), which is widely used as $\mathcal{L}_{im} = \mathcal{L}_{ent} + \mathcal{L}_{div} = \mathbb{E}_{x_T \in \mathcal{X}_T} H(f_T(x_T)) - H(\bar{f}_T(x_T))$, where $H$ denotes a conditional entropy function, and $\bar{f}_T(x_T) = \mathbb{E}_{x_T \in \mathcal{X}_T} f_T(x_T)$.

Taken together, the final objective of the target model is given by $\mathcal{L}_t = \lambda_t \mathcal{L}_{sl} + \mathcal{L}_{im}$, where $\lambda_t = \exp(-I/I_{target})$ is a hyper-parameter with the exponential decay with respect to iteration $I$. $I_{target}$ is a total training iteration of target adaptation. During the target adaptation, the label refinement phase is performed at regular intervals.

## 4 Experiments

### 4.1 Experimental setting

**Datasets, training parameters, and implementation details.** We evaluate the performance of the proposed method on four benchmark datasets, *i.e.*, Office (Hoffman et al., 2018b), Office-Caltech (Saenko et al., 2010), Office-Home (Venkateswara et al., 2017) and DomainNet (Peng et al., 2019), that include data samples in different domains. We designate one of the domains as a target domain and the remaining domains as source domains as described in Sec. E.1.

For fair comparisons, we follow the same experimental settings as previous works (Liang et al., 2020; Ahmed et al., 2021; Liang et al., 2022). We describe a detailed training parameter setting such as training epochs, learning rates, and batch sizes in Sec. E.1.

We use two deep models with ResNet101 (He et al., 2016) and ViT-B_16 (ViT16 for simplicity) (Dosovitskiy et al.) for source models. ResNet-101 is used as the target model. PRN is trained in every epoch of a target model training, whereas the warm-up phase is conducted once. We have reinitialized a learning rate at each training epoch of the refinement phase for reliable training, when $p_c$ and $p_d$ change on the same sample. The total training epoch is 15. We measure the performance of the proposed method three times using different random seeds $\{2019, 2020, 2021\}$ via PyTorch (Paszke et al., 2017) and report the average performances.

**Performance Comparison.** We compare our method with state-of-the-art MSDA methods to evaluate its effectiveness. **No Adapt.** (also known as "Source Only") denotes the test performance on target data, when a model is trained using only the source data. Considering the MS setting, we further compare **No Adapt (SB)** that is assumed to choose the most suitable source prediction for the adaption, which is the ideal scenario but usually infeasible. In contrast, **No Adapt (SW)** is assumed that a user chooses the worst source prediction. Moreover, we provide **No Adapt (MS)** when the source model is trained using all the source data from available source domains.

We categorize the tested DA methods with their MS settings as MSDA, MSFDA, and MSBDA. For MSDA, we present the adaptation results from $\mathbf{M^3SDA}$, $\mathbf{M^3SDA}$-$\beta$ (Peng et al., 2019), $\mathbf{SImpAI}_{101}$ (Venkat et al., 2020), and **MFSAN** (Zhu et al., 2019). Both source data and models are available in these methods. Furthermore, we compare MSFDA methods, including **SHOT**, **SHOT++** (Liang et al., 2021), **DECISION** (Ahmed et al., 2021), and **CAiDA** (Dong et al., 2021). For (Liang et al., 2021), we also report the performance of **SHOT-ens**, which takes an average of the soft prediction as a target prediction, to pose the method in an MS setting. **DINE** (Liang et al., 2022) is a single-source BDA method and extended to MSBDA, thus used for performance comparisons. DINE used a two-step learning procedure, including source adaptation to a target model with **DINE (w/o FT)** and further fitting to a specific target domain with **DINE (FT)**. Because the primary objective of a generic UDA is to generalize target domains and maintain its performance even when faced with different domain shifts, we mainly compared the performance of DINE (w/o FT) with our method.

## 4.2 EXPERIMENTAL RESULTS

The classification results on Office, Office-Caltech, Office-Home, and DomainNet are shown in Table 1–2. "MU", "MS", and "MB" refer to MSDA, MS-SFDA, and MSBDA, respectively. A right arrow "→" refers to an adaptation task. For instance, "→ W" denotes the target adaptation from all the other domains to a domain "W." Because MSDA and MS-SFDA have access to model parameters, the source and target models have the same structure, and they are tested, only when the models use ResNet101 in common. BDA does not have such constraints, allowing for ViT16 as a source model.

In the MSBDA setting, our method achieves competitive accuracy across four datasets, and notably, it demonstrates comparable performance to the other methods in the MSDA and MS-SFDA settings. Our method accomplishes these results even in the absence of source data and models. The highest accuracy achieved within the MSBDA setting is highlighted in **bold**. Compared to DINE, the proposed method consistently provides superior classification performance across all datasets and source models. This result demonstrates the efficiency of our approach in transferring positive knowledge from source domains without the aid of source data and models.

**Results on Office.** We present the adaptation results on Office in Table 1. Our method outperformed DINE (w/o FT) with a margin of 3.6% on average. Furthermore, our method achieved comparable performances to UDA and SFDA methods. Compared to CAiDA (Dong et al., 2021), which is the state-of-the-art MS-SFDA method, our approach achieved an increase of 0.2% with ResNet101.

**Results on Office-Caltech.** Office-Caltech has been known to be comparably easy for target adaptation, when considering its accuracy in "No Adapt". Most of the MSDA and MS-SFDA methods achieved substantial performance improvements as compared to "No Adapt (MS)". Our method outperformed DINE (w/o FT) by 1.7% in ResNet101.

**Results on Office-Home.** Our method exhibited similar phenomenon on Office-Home. The proposed method achieved the best accuracy for all the tasks in UDA and SFDA methods. LPR also outperformed DINE (w/o FT) by the margin of 4.2% on average.

**Results on DomainNet.** Table 2 displayed the performance on DomainNet, which was challenging task due to a large number of categories and their associated large discrepancies. Compared to other

Table 1: Classification accuracy (%) on Office, Office-Caltech, and Office-Home.

| $f_S$ | Methods | Setting | Office →A | →D | →W | Avg. | Office-Caltech →A | →C | →D | →W | Avg. | Office-Home →Ar | →Cl | →Pr | →Re | Avg. |
|---|---|---|---|---|---|---|---|---|---|---|---|---|---|---|---|---|
| | No Adapt (SB) | - | 64.8 | 98.2 | 94.8 | 86.0 | 95.6 | 90.2 | 100.0 | 95.9 | 95.4 | 69.1 | 50.1 | 79.9 | 76.7 | 68.9 |
| | No Adapt (SW) | - | 53.9 | 81.5 | 81.1 | 72.2 | 84.1 | 81.5 | 95.5 | 90.5 | 87.9 | 55.0 | 44.7 | 66.0 | 68.4 | 58.5 |
| | No Adapt (MS) | - | 64.5 | 82.3 | 80.7 | 75.8 | 84.9 | 88.7 | 93.0 | 88.5 | 88.8 | 54.9 | 49.9 | 69.6 | 76.7 | 62.8 |
| ResNet101 | SImpAI$_{101}$ (Venkat et al., 2020) | MU | 99.4 | 97.9 | 71.2 | 89.5 | 100. | 100. | 94.6 | 95.6 | 97.5 | 73.4 | 62.4 | 81.0 | 82.7 | 74.8 |
| | MFSAN (Zhu et al., 2019) | MU | 72.7 | 99.5 | 98.5 | 90.2 | - | - | - | - | - | 72.1 | 62.0 | 80.3 | 81.8 | 74.1 |
| | SHOT (Liang et al., 2021) | MS | - | - | - | - | 96.2 | 96.2 | 98.5 | 99.8 | 97.7 | 73.0 | 60.4 | 83.9 | 83.3 | 75.2 |
| | DECISION (Ahmed et al., 2021) | MS | 75.4 | 98.4 | 99.6 | 91.1 | 95.9 | 95.9 | 100. | 99.6 | 97.9 | 74.5 | 59.4 | 84.4 | 83.6 | 75.5 |
| | SHOT ++ (Liang et al., 2021) | MS | - | - | - | - | 96.2 | 96.5 | 99.4 | 100. | 98.0 | 73.1 | 61.3 | 84.3 | 84.0 | 75.7 |
| | CAiDA (Dong et al., 2021) | MS | 75.8 | 99.8 | 98.9 | 91.6 | 96.8 | 97.1 | 100. | 99.8 | 98.4 | 75.2 | 60.5 | 84.7 | 84.2 | 76.2 |
| | DINE (w/o FT) (Liang et al., 2022) | MB | 69.2 | 98.6 | 96.9 | 88.2 | 95.0 | 92.0 | 98.5 | 97.3 | 95.7 | 70.8 | 57.1 | 80.9 | 82.1 | 72.7 |
| | DINE (FT) (Liang et al., 2022) | MB | 76.8 | 99.2 | 98.4 | 91.5 | 95.9 | 95.2 | 98.9 | 97.1 | 96.8 | 74.5 | 64.1 | 85.0 | 84.6 | 77.1 |
| | **LPR (Ours)** | MB | 77.2 | 99.5 | 98.7 | 91.8 | 95.9 | 95.3 | 98.6 | 99.9 | 97.4 | 74.8 | 64.5 | 85.8 | 84.8 | 77.5 |
| ViT16 | No Adapt (SB) | - | 71.9 | 99.4 | 96.9 | 89.4 | 96.0 | 93.5 | 99.4 | 99.7 | 97.1 | 79.6 | 54.7 | 88.8 | 87.2 | 77.6 |
| | No Adapt (SW) | - | 70.9 | 84.5 | 84.7 | 80.0 | 88.5 | 84.1 | 94.3 | 92.2 | 89.8 | 74.4 | 50.9 | 83.0 | 85.5 | 73.5 |
| | No Adapt (MS) | - | 77.2 | 88.2 | 89.2 | 84.9 | 92.9 | 95.9 | 98.7 | 95.8 | 95.8 | 74.5 | 54.5 | 83.2 | 87.2 | 74.9 |
| | DINE (w/o FT) (Liang et al., 2022) | MB | 80.7 | 98.4 | 97.1 | 92.1 | 96.4 | 96.0 | 99.4 | 98.2 | 97.5 | 82.4 | 61.0 | 88.6 | 90.8 | 80.7 |
| | DINE (FT) (Liang et al., 2022) | MB | 82.4 | 99.2 | 98.4 | 93.3 | 96.8 | 97.0 | 99.6 | 98.8 | 98.1 | 83.6 | 67.0 | 90.9 | 91.8 | 83.3 |
| | **LPR (Ours)** | MB | 82.6 | 99.5 | 98.3 | 93.4 | 96.9 | 97.2 | 99.5 | 99.3 | 98.2 | 83.3 | 67.2 | 92.3 | 93.5 | 84.0 |

Table 2: Classification accuracy (%) on DomainNet.

| $f_S$ | Methods | Setting | →Clp | →Inf | →Pnt | →Qdr | →Rel | →Skt | Avg. |
|---|---|---|---|---|---|---|---|---|---|
| ResNet101 | No Adapt (SB) | - | 52.4 | 20.5 | 48.4 | 13.5 | 58.1 | 43.7 | 39.4 |
| | No Adapt (SW)* | - | 10.6 | 1.2 | 1.9 | 2.7 | 4.4 | 9.2 | 5.0 |
| | No Adapt (MS)* | - | 47.6 | 13.0 | 38.1 | 13.3 | 51.9 | 33.7 | 32.9 |
| | M$^3$SDA (Peng et al., 2019) | MU | 57.2 | 24.2 | 51.6 | 5.2 | 61.6 | 49.6 | 41.5 |
| | M$^3$SDA-$\beta$ (Peng et al., 2019) | MU | 58.6 | 26.0 | 52.3 | 6.3 | 62.7 | 49.5 | 42.6 |
| | SImpAI$_{101}$ (Venkat et al., 2020) | MU | 66.4 | 26.5 | 56.6 | 18.9 | 68.0 | 55.5 | 48.6 |
| | SHOT-ens (Liang et al., 2021) | MS | 58.6 | 25.2 | 55.3 | 15.3 | 70.5 | 52.4 | 46.2 |
| | DECISION (Ahmed et al., 2021) | MS | 63.2 | 22.3 | 54.6 | 18.2 | 67.9 | 51.4 | 46.3 |
| | DINE (w/o FT) (Liang et al., 2022)* | MB | 61.4 | 21.5 | 54.7 | 13.4 | 70.9 | 50.3 | 45.4 |
| | DINE (FT) (Liang et al., 2022)* | MB | 55.6 | 6.3 | 33.6 | 0.2 | 25.7 | 32.1 | 25.6 |
| | **LPR (Ours)** | MB | 63.9 | 24.0 | 55.1 | 13.6 | 70.8 | 51.2 | 46.4 |
| ViT16 | No Adapt (SB)* | - | 55.2 | 19.9 | 47.0 | 14.9 | 62.1 | 45.2 | 40.7 |
| | No Adapt (SW)* | - | 4.5 | 0.5 | 0.9 | 4.1 | 1.6 | 6.4 | 3.0 |
| | DINE (w/o FT)(Liang et al., 2022)* | MB | 63.0 | 21.6 | 56.7 | 15.0 | 72.8 | 50.2 | 46.5 |
| | DINE (FT) (Liang et al., 2022)* | MB | 58.2 | 5.3 | 35.9 | 0.2 | 24.8 | 33.5 | 26.3 |
| | **LPR (Ours)** | MB | 65.2 | 24.2 | 56.9 | 15.8 | 73.1 | 50.6 | 47.6 |

datasets, it is hard to achieve high adaptation accuracy in every setting. Nevertheless, our method achieved comparable performance to all the methods except SImpAI$_{101}$ (Venkat et al., 2020).

DINE has significantly degraded performance by approximately 20%, when it used the FT in DomainNet. In (Liang et al., 2022), the FT has been optimized to fit the Office datasets and improved the performance in the same datasets. However, DomainNet dataset was not used for the optimization, and the FT could worsen the domain shift problem. Our method did not experience such failure.

## 4.3 PERFORMANCE ANALYSIS AND ABLATION STUDIES

Table 3: Ablation studies on Office, (a) when each component including warm up (WU), label refinement (LR), and target adaptation (TA) is turned on or off, and (b) when the loss functions are differently combined during the LR phase.

(a)

| WU | LR | TA | →A | →D | →W | Avg. |
|---|---|---|---|---|---|---|
| | | ✓ | 62.4 | 89.9 | 89.5 | 80.6 |
| ✓ | | ✓ | 57.8 | 89.4 | 87.6 | 78.3 |
| | ✓ | ✓ | 75.6 | 88.9 | 97.1 | 87.2 |
| ✓ | ✓ | ✓ | 77.2 | 99.5 | 98.7 | 91.8 |

(b)

| $\mathcal{L}_{cc}$ | $\mathcal{L}_{cd}$ | $\mathcal{L}_{ld}$ | $\mathcal{L}_s$ | →A | →D | →W | Avg. |
|---|---|---|---|---|---|---|---|
| ✓ | | | | 71.0 | 97.2 | 95.4 | 87.8 |
| ✓ | ✓ | | | 72.0 | 97.6 | 95.7 | 88.4 |
| ✓ | ✓ | ✓ | | 73.6 | 98.7 | 96.8 | 89.7 |
| ✓ | ✓ | ✓ | ✓ | 77.2 | 99.5 | 98.7 | 91.8 |

**Effect of phases.** In Table 3a, the warm-up (WU) and label refinement (LR) steps of the PRN significantly improved the adaptation performance. LR has improved 6.6% in comparison to the TA only. WU further improved 4.6 % . The tests demonstrated the efficacy of the components.

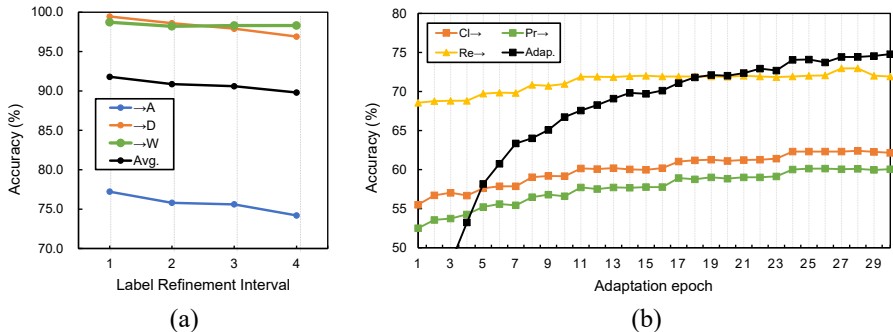

Figure 3: (a) Ablation studies on PRN with loss functions and (b) a refinement interval. (a) and (b) are tested with an Office dataset. (c) Classification accuracy of the refined predictions and target prediction during the target adaptation phase for "→ Ar" task on an Office-Home dataset.

**Loss function.** In Table 3b, we presented the adaptation accuracy during LR, when each loss in Eq.(11) was added. ResNet101 was used for both $f_S$ and $f_T$. LPR achieved a similar performance to DINE (w/o FT), when using only the $\mathcal{L}_{cc}$. The performance has been further improved using $\mathcal{L}_{cd}$ and $\mathcal{L}_{ld}$ by the margin of 0.6% and 1.3%, respectively. We achieved further improvements of 2.1% with $\mathcal{L}_s$. More ablation tests are presented with various $\lambda_{cd}$, $\lambda_{ld}$ and $\lambda_s$ of Eq. (11) in Sec E.4.

**Training interval of LR.** We then analyze the training interval of a LR phase. The LR phase was performed every 1 epoch of target adaptation. In Figure 3(a), we illustrated the adaptation performance, when the interval increased to 2, 3, and 4 epochs. We found that 1 epoch was appropriate and the performance of the adaptation would degrade if the refinement process was not sufficiently applied. In other words, the refinement process was effective during adaptation.

**Investigation of a domain adaptation phase.** In Figure 3(b), we illustrated how the quality of refined source predictions changes during the target adaptation. We reported the result of the adaptation to Art (Ar) domain of Office-Home. The orange, green and yellow curve displayed the variations of the quality of source predictions from Cl, Pr, and Re source, respectively, and the black curve displayed the adaptation performance. In the beginning, each source curve presented the quality of the output of the PRN after the warm-up. Up to 2 training epochs, the quality tended to decrease, because the target predictions up to this point were not helpful for refinement. However, the quality constantly increased as the refinement phase went on, and the accuracy kept improving. Noisy source predictions illustrated as orange and green curves have been further refined with a larger gradient than the yellow curve. There were more noisy samples affected by the refinement phase, and the quality has been improved, which implies the phase has been effective.

**Selection of representative prediction $p_c$ and $p_d$.** In Sec. 3.2, we derive the condition of an optimal $p_c$. Nevertheless, because $y_T$ is unknown, it is intractable to precisely determine the representative predictions. We presented a method to designate $p_c$ in Sec. 3.3, allowing for leveraging complementary information from many sources to avoid the incorrect designation. To ensure the effectiveness, we test several alternatives including a random selection (RS), the selection of a confident prediction (CP). The proposed method presented a superior performance to the alternatives, empirically justifying the selection of $p_c$. We also chose $p_d$ to have lower confidence prediction in $\mathcal{P}_d$ and test several alternatives to justify the performance. The results can be found in Sec. E.5.

## 5 CONCLUSION AND FUTURE WORK

In this paper, we proposed a novel MSBDA pseudo-labeling framework that used only the predictions of source models to explore positive knowledge from multiple source domains. We derived a theoretical analysis to demonstrate the effectiveness of the proposed training method and justified the design of the PRN architecture to use an attention mechanism to resolve complex relations among different domains. We evaluated the performance of the proposed method on various benchmark datasets and demonstrated its superiority in comparison to state-of-the-art methods in various domain adaptation settings. In the future work, we will investigate open-set MSBDA problems.

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

SUPPLEMENTARY MATERIAL

## A  PROOF OF THEOREMS

We define the triangular inequality for errors in (Crammer et al., 2008) stated below.

**Definition 1. (Triangular inequality for errors)** *For any hypothesis $f_1, f_2, f_3$ in class $\mathcal{H}$,*

$$\epsilon(f_1, f_2) \leq \epsilon(f_1, f_3) + \epsilon(f_2, f_3). \tag{12}$$

**Theorem 1. (General upper bound of a risk in target prediction)** *Denote $h$ as a hypothesis in $\mathcal{H}$. We then establish a theoretical upper bound on the difference between the target error and the oracle error as*

$$|\epsilon(h, \hat{p}) - \epsilon(h, y_T)| \leq \sum_i \alpha_i \epsilon(p_i, y_T), \tag{13}$$

*where a pseudo label is defined as $\hat{p} = \sum_i \alpha_i p_i$, $\alpha_i \geq 0$, $\sum_i \alpha_i = 1$.*

*Proof:*

First, we find the upper bound of the target error.

$$\epsilon(h, \hat{p}) = \mathbb{E}_{x \in \mathcal{D}_t}\left[|h(x) - \hat{p}|\right] \tag{14}$$

$$= \mathbb{E}_{x \in \mathcal{D}_t}\left[\left|h(x) - \sum_{i=1}^{M} \alpha_i p_i\right|\right] = \mathbb{E}_{x \in \mathcal{D}_t}\left[\left|\sum_{i=1}^{M} \alpha_i(h(x) - p_i)\right|\right] \tag{15}$$

$$\leq \sum_{i=1}^{M} \alpha_i \mathbb{E}_{x \in \mathcal{D}_t}\left[|h(x) - p_i|\right] = \sum_{i=1}^{M} \alpha_i \epsilon(h, p_i). \tag{16}$$

Then, we find the upper bound by applying **Definition 1**.

$$|\epsilon(h, \hat{p}) - \epsilon(h, y_T)| \tag{17}$$

$$\leq \left|\sum_{i=1}^{M} \alpha_i \epsilon(h, p_i) - \epsilon(h, y_T)\right| \tag{18}$$

$$\leq \sum_{i=1}^{M} \alpha_i |\epsilon(h, p_i) - \epsilon(h, y_T)| \tag{19}$$

$$\leq \sum_{i=1}^{M} \alpha_i \epsilon(y_T, p_i). \tag{20}$$

□

**Lemma 1. (Modified upper bound of a risk)**

$$|\epsilon(h, \hat{p}) - \epsilon(h, y_T)| \leq \epsilon(p_c, y_T) + \eta, \tag{21}$$

*where $p_c = \arg\min_{p_i \in \mathcal{P}} \epsilon(p_i, y_T)$, and $\eta = \sum_i \alpha_i \epsilon(p_c, p_i)$.*

*Proof:*

Applying triangular inequality to each pseudo error,

$$|\epsilon(h, \hat{p}) - \epsilon(h, y_T)| \tag{22}$$

$$\leq \sum_{i=1}^{M} \alpha_i |\epsilon(y_T, p_c) + \epsilon(p_c, p_i)| \tag{23}$$

$$\leq \epsilon(y_T, p_c) + \sum_{i=1}^{M} \alpha_i \epsilon(p_c, p_i) = \epsilon(p_c, y_T) + \eta. \tag{24}$$

□

Let us assume there exist $\mathcal{P}_c$ and $\mathcal{P}_d \subset \mathcal{P}$ as two subsets of an entire label space. $\mathcal{P}_c$ and $\mathcal{P}_d$ are defined as the spaces, in which their samples are concentrated to $p_c$ and dispersed from $p_c$, respectively. They are mathematically defined as follows:

$$\mathcal{P}_c = \{p_i | \epsilon(p_c, p_i) \leq \xi\}, \quad \mathcal{P}_d = \mathcal{P} \setminus \mathcal{P}_c, \tag{25}$$

where $\xi$ denotes a threshold for the label space division.

**Theorem 2. (Upper bound of a risk with a label space division)**

$$|\epsilon(h, \hat{p}) - \epsilon(h, y_T)| \leq \epsilon(p_c, y_T) + \eta_c + \eta_d + \sum_{p_i \in \mathcal{P}_d} \alpha_i \epsilon(p_c, p_d), \tag{26}$$

*where $p_d = \arg\min_{p_i \in \mathcal{P}_d} \epsilon(p_c, p_i)$ is the representative prediction in $\mathcal{P}_d$. $\eta_c = \sum_{p_i \in \mathcal{P}_c} \alpha_i \epsilon(p_c, p_i)$ and $\eta_d = \sum_{p_i \in \mathcal{P}_d} \alpha_i \epsilon(p_d, p_i)$ are the dispersion of $\mathcal{P}_c$ and $\mathcal{P}_d$, respectively.*
*Proof:*

We apply the triangular inequality to the terms related to concentration.

$$|\epsilon(h, \hat{p}) - \epsilon(h, y_T)| \tag{27}$$

$$\leq \epsilon(p_c, y_T) + \sum_{p_i \in \mathcal{P}_c} \alpha_i \epsilon(p_c, p_i) + \sum_{p_i \in \mathcal{P}_d} \alpha_i \epsilon(p_c, p_i) \tag{28}$$

$$\leq \epsilon(p_c, y_T) + \eta_c + \sum_{p_i \in \mathcal{P}_d} [\alpha_i \epsilon(p_c, p_d) + \alpha_i \epsilon(p_d, p_i)] = \epsilon(p_c, y_T) + \eta_c + \eta_d + \sum_{p_i \in \mathcal{P}_d} \alpha_i \epsilon(p_c, p_d). \tag{29}$$

$\square$

## B  NOTATIONS

We summarize all the notations of variables in Table 4.

## C  ARCHITECTURE OF PRN

We provide a detailed architecture of the proposed pseudo label refinery network in Table 5. PRN is fed by the concatenated source prediction $\mathcal{P} \in \mathcal{R}^{M \times K}$ and outputs the refined predictions with the same size of the input, $\mathcal{P}^w$ and $\mathcal{P}^r$ in the warm-up and adaptation phase, respectively. The learnable parameters are only related to the fully connected layer and the total number is $NK^2$.

## D  PSEUDO CODE

We present the overall procedure of the proposed MSBDA training framework in Algorithm 1. All the variables are defined in Table 4.

## E  EXPERIMENTAL DETAILS

### E.1  EXPERIMENTAL RESOURCES AND IMPLEMENTATION DETAILS

#### E.1.1  DATASETS

We evaluate the proposed method on four benchmark datasets, including Office (Hoffman et al., 2018b), Office-Caltech (Saenko et al., 2010), Office-Home (Venkateswara et al., 2017) and DomainNet (Peng et al., 2019). Office dataset consists of three domains, *i.e.*, Amazon (A), DSLR (D), and Web (W) with 31 categories. Office-Caltech dataset is an extension of the Office dataset with 10 categories in Amazon (A), CalTech (C), DSLR (D), and Web (W) domains. Office-Home dataset contains four domains, *i.e.*, Art (Ar), Clipart (Cl), Product (Pr), and Real-Word (Rw) with 65

Table 4: Definition of the notations.

| Variables | Definition |
|---|---|
| $M$ | Number of source domains |
| $K$ | Number of classification categories |
| $\mathcal{D}_i, 1 \le i \le M$ | $i$-th source domain |
| $\mathcal{D}_s$ | Collection of source domains |
| $\mathcal{D}_T$ | Target domain |
| $\mathcal{X}, \mathcal{Y}$ | Set of samples and annotations |
| $x_T, y_T$ | Target sample and its groundtruth |
| $f_i$ | Source model pre-trained using $\mathcal{D}_i$ |
| $f_t$ | Target model |
| PRN | Pseudo label refinery network |
| $N$ | Number of refinement blocks of PRN |
| $p_i$ | Source prediction on target sample by $f_i$ |
| $\mathcal{P} \in \mathbb{R}^{M \times K}$ | Set of source predictions |
| $p_c$ | Representative label in $\mathcal{P}$ |
| $\mathcal{P}_c \in \mathcal{P}$ | Set of predictions concentrated to $p_c$ |
| $\mathcal{P}_d = \mathcal{P} \setminus \mathcal{P}_c$ | Set of predictions dispersed from $p_c$ |
| $p_d$ | Representative label in $\mathcal{P}_d$ |
| $I_{\text{source}}$ | Training iteration of source models |
| $I_{\text{warmup}}, I_{\text{refine}}, I_{\text{target}}$ | Training iteration of warm-up, label refinement, and target adaptation phase |
| $T_{\text{refine}}$ | Training interval of label refinement phase |
| $\eta_{\text{warmup}}, \eta_{\text{refine}}, \eta_{\text{target}}$ | Initial learning rate of warm-up, label refinement, and target adaptation phase |
| $\lambda_{cd}, \lambda_{ld}, \lambda_s$ | Hyperparameter in $\mathcal{L}_{cd}, \mathcal{L}_{ld}, \mathcal{L}_s$ |
| $B$ | Batch size |

Table 5: Detailed architecture of PRN.

| Block name | Output size | Architecture | |
|---|---|---|---|
| Refinement block | $M \times K$ | Attention layer (AT) 
 Fully connected layer (FC) 
 ReLU 
 Dropout 
 Residual connection 
 Normalization | $\times N$ |
| Softmax | $M \times K$ | | |

categories. DomainNet dataset, which is considered as the most challenging one in MSDA, includes approximately 0.6 million images and 345 categories in six domains, *i.e.*, Clipart (Clp), Infograph (Inf), Painting (Pnt), Quickdraw (Qdr), Real (Rel), and Sketch (Skt). We designate one of the domains as the target domain and the remaining domains as the source domains.

E.1.2 TRAINING AND IMPLEMENTATION DETAILS

For fair comparisons, we follow the same experimental settings as previous works (Liang et al., 2020; Ahmed et al., 2021; Liang et al., 2022). The network is trained with a momentum value of 0.9 and a weight decay of $10^{-3}$ in a stochastic gradient descent (SGD) optimizer and the same learning rate scheduling strategy as in (Ahmed et al., 2021). The batch size is set to 64 for Office, Office-Caltech, and Office-Home and 32 for DomainNet.

All the experiments on Office (Hoffman et al., 2018b), Office-Caltech (Saenko et al., 2010) and Office-Home (Venkateswara et al., 2017) were conducted using 1 12GB Geforce GTX TITAN X. A single 48GB NVIDIA RTX A6000 was used for the experiments on DomainNet (Peng et al., 2019).

We use two deep models with ResNet101 (He et al., 2016) and ViT-B_16 (ViT16 for simplicity) (Dosovitskiy et al.) for source models. They have been pre-trained on the ImageNet (Deng et al.,

---

**Algorithm 1** Training of PRN and target model

---

**Input:** $\mathcal{D}_T$, $\mathcal{P}$, $I_{\text{warmup}}$, $I_{\text{refine}}$, $I_{\text{target}}$, $T_{\text{refine}}$.
**Output:** well-trained PRN and $f_T$.
**for** $iter = 1$ **to** $I_{\text{warmup}}$ **do**
   Train PRN according to Section 3.4.1.
**end for**
**for** $iter = 1$ **to** $I_{\text{target}}$ **do**
   Train $f_T$ according to Section 3.4.3.
   **if** $iter\%T_{\text{refine}} == 0$ **then**
      Divide the label space according to Section 3.3.
      **for** $iter = 1$ **to** $I_{\text{refine}}$ **do**
         Train PRN according to Section 3.4.2.
      **end for**
   **end if**
**end for**

---

2009) and fine-tuned on source samples. The number of training epochs is set to 100, 100, 50, and 50 for Office, Office-Caltech, Office-Home, and DomainNet datasets, respectively (Ahmed et al., 2021). ResNet-101 is used as the target model. During the warm-up phase of PRN, the learning rate is set to $10^{-3}$ for every dataset, and the maximum training epoch is set to one. During the adaptation phase, the initial learning rate is set to $10^{-3}$ for the DomainNet dataset and $10^{-4}$ for the other datasets.

### E.2 TRAINING PARAMETERS

We show the detailed parameter values set to the experiments on different databases in Table 6.

Table 6: Parameter settings on different databases. $I$ and $T$ are the number of epochs.

| Parameter | Office | Office-Caltech | Office-Home | DomainNet |
|---|---|---|---|---|
| $N$ | 2 | 2 | 2 | 1 |
| $I_{\text{source}}$ | 100 | 100 | 50 | 50 |
| $I_{\text{warmup}}$ | 1 | 1 | 1 | 1 |
| $I_{\text{refine}}$ | 1 | 1 | 1 | 1 |
| $I_{\text{target}}$ | 30 | 30 | 30 | 15 |
| $T_{\text{refine}}$ | 1 | 1 | 1 | 1 |
| $\eta_{\text{warmup}}$ | $10^{-3}$ | $10^{-3}$ | $10^{-3}$ | $10^{-3}$ |
| $\eta_{\text{refine}}$ | $10^{-4}$ | $10^{-4}$ | $10^{-4}$ | $10^{-3}$ |
| $\eta_{\text{target}}$ | $10^{-4}$ | $10^{-4}$ | $10^{-4}$ | $10^{-3}$ |
| $\lambda_{\text{cd}}$ | 1 | 1 | 1 | 1 |
| $\lambda_{\text{ld}}$ | 1 | 1 | 1 | 1 |
| $\lambda_s$ | 1 | 1 | 1 | 1 |
| $B$ | 64 | 64 | 64 | 32 |

We used the same parameter values of $I_{source}$, $I_{target}$ and $\eta_{target}$ in Table 6, as in the previous studies (Ahmed et al., 2021; Liang et al., 2022), because these parameters contribute to the training of the target model itself. The primary goal of this work was the investigation of selective knowledge transfer in MSBDA. Notably, for fair comparisons, the PRN is independently trained, ensuring that parameter selection has minimal impact, irrespective of the proposed modules. Thus, we have not applied any selection methods and used the same hyper-parameters as the previous studies.

Concerning parameters related to the training of PRN, we conducted a grid search for the hyperparameters of $\eta_{warmup}$, $\eta_{refine}$, $\lambda_{cd}$, $\lambda_{ld}$ and $\lambda_s$. The selected values for these parameters are given in Table 6. The sensitivity analysis of $\lambda_{cd}$, $\lambda_{ld}$ and $\lambda_s$ are shown in Sec. E.4.

### E.3 STANDARD DEVIATIONS

Since we performed the experiments on the proposed method three times with different random seeds, we displayed the standard deviations of the experimental results in Table 7 and 8. The standard deviations on all the tasks range from 0.05 to 1.03.

Table 7: Standard deviations of the experiments on Office, Office-Caltech, and Office-Home.

| $f_S$ | Office | | | | Office-Caltech | | | | | Office-Home | | | | |
|---|---|---|---|---|---|---|---|---|---|---|---|---|---|---|
| | →A | →D | →W | Avg. | →A | →C | →D | →W | Avg. | →Ar | →Cl | →Pr | →Re | Avg. |
| ResNet101 | 0.64 | 0.38 | 0.21 | 0.13 | 0.08 | 0.41 | 0.19 | 0.16 | 0.13 | 0.05 | 0.34 | 0.27 | 0.23 | 0.22 |
| ViT16 | 0.05 | 0.09 | 0.12 | 0.05 | 0.07 | 0.08 | 0.06 | 0.08 | 0.07 | 0.25 | 0.53 | 0.28 | 0.27 | 0.06 |

Table 8: Standard deviations of the experiments on DomainNet.

| $f_S$ | →Clp | →Inf | →Pnt | →Qdr | →Rel | →Skt | Avg. |
|---|---|---|---|---|---|---|---|
| ResNet101 | 0.37 | 0.10 | 0.25 | 0.22 | 0.40 | 0.27 | 0.03 |
| ViT16 | 0.12 | 0.53 | 0.34 | 1.03 | 0.20 | 0.54 | 0.46 |

### E.4 SENSITIVITY ANALYSIS OF HYPERPARAMETERS

We conducted ablation studies on the hyperparameters $\lambda_{cd}$, $\lambda_{ld}$, and $\lambda_s$ used in the label refinement phase on Office. We changed the values of these parameters in the range of $[0, 1]$ and analyzed the effects of the parameters on the adaptation. In Figure 4 (a)-(c), we observed that changing the parameters did not have a significant impact on the performance in our experiment. Specifically, the adaptation performances were stable in the range of [0.4, 1], [0.4, 1] and [0.6, 1] of $\lambda_{cd}$, $\lambda_{ld}$, and $\lambda_s$, respectively.

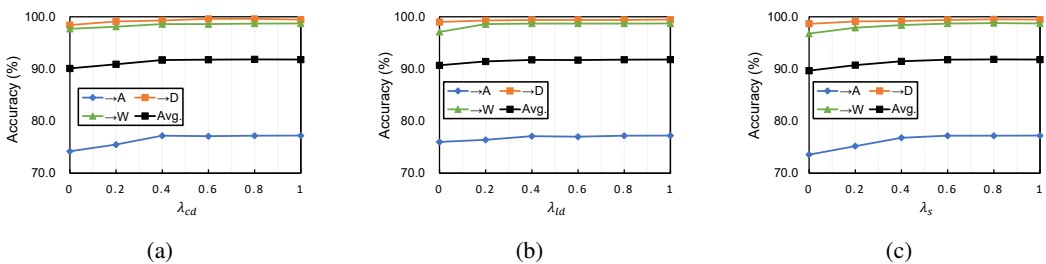

(a)       (b)       (c)

Figure 4: Sensitivity analysis of (a) $\lambda_{cd}$, (b) $\lambda_{ld}$ and (c) $\lambda_s$

### E.5 SELECTION RULE OF $p_c$ AND $p_d$

An issue on the theoretical analysis in Section 3.2 is that one cannot find the representative prediction $p_c = \arg\min_{p_i \in \mathcal{P}} \epsilon(p_i, y_T)$ since $y_T$ is unknown in the unsupervised setting on the target domain. Instead, we presented a practical method on how to designate $p_c$ in Section 3.3. The intuition of such designation is to leverage complementary information from as many sources as possible to avoid the incorrect designation of $p_c$. To ensure the effectiveness of the proposed method, we test several alternatives. In random selection (RS), we randomly select a prediction as $p_c$ and define $\mathcal{P}_c$ with the predictions classifying the same category of $p_c$. $\mathcal{P}_d$ are the set of the rest predictions. In confident prediction (CP), we first pick the prediction with the highest probability as $p_c$, which does not consider the categorical agreement between predictions. The definition of $\mathcal{P}_c$ and $\mathcal{P}_d$ is the same as random selection.

Table 9a displayed the results when using different methods on choosing $p_c$. The experiments are on Office-Home database and $f_S$ is ResNet101.

RS has substantially degraded the performance of the proposed method. In RS, $p_c$ tends to be incorrect in the beginning. Once an undesirable prediction is designated as $p_c$, the training objectives $\mathcal{L}_{cd}$ and $\mathcal{L}_{ld}$ cause predictions in $\mathcal{P}_d$ to become noisy as well thus producing pseudo labels in low quality.

CP overcame RS and improved the adaptation performance. However, it showed lower performance than our method, implying that the prediction with the highest probability is not always best option for $p_c$. Even if source prediction has a high probability on a certain category, it does not guarantee the prediction is the best choice since predictions are inherently noisy due to the distributional shift between source and target.

Our method, which aggregates the majority of the predictions, presented the best adaptation results among all the methods. This method solves some failure cases in the confident prediction by providing additional helpful support on defining $p_c$. Even though the confidence of source predictions is low with low probability, an agreement between predictions can act as a criterion of defining $\mathcal{P}_c$. The improvement in results to the confident prediction implies that the agreement takes precedence over the confidence of each prediction.

**Selection of representative prediction $p_d$.** Once $\mathcal{P}_d$ is defined as in Section 3.3, we define the representative label $p_d$ in $\mathcal{P}_d$ as the prediction whose highest probability is minimum among the remaining predictions in $\mathcal{P}_d$. This is because the prediction with lower confidence is more likely to be a $p_d = \arg\min_{p_i \in \mathcal{P}_d} \epsilon(p_c, p_i)$ as defined in Theorem 2. Intuitively speaking, such unconfident prediction is liable to change its classification and tends to locate closer to $p_c$ by $\mathcal{L}_{ld}$ during the label refinement phase. To verify the effective selection of $p_d$, we tested different methods. Random selection (RS) refers to selecting random $p_d$ in $\mathcal{P}_d$. In Maximum probability (MaxP), we select the prediction with the highest probability as $p_d$.

In Table 9b, we compared the different methods for defining $p_d$. The selection method of $p_c$ was set as explained in Section 3.3 for all the tests. RS and MaxP exhibited similar adaptation performances on all the tasks. The proposed method outperformed both RS and MaxP on every task, implying that prediction with the minimum highest probability is a better choice for $p_d$.

Table 9: Various methods to choose (a) $p_c$ and (b)$p_d$ and their performance. RS refers to random selection, CP refers to the solution of the highest probability without categorical agreement, and MaxP refers to the solution of the highest probability in $\mathcal{P}_d$.

(a)

| Methods | →Ar | →Cl | →Pr | →Re | Avg. |
|---------|------|------|------|------|------|
| RS | 61.2 | 50.8 | 75.8 | 73.9 | 65.4 |
| CP | 68.2 | 57.1 | 79.4 | 78.2 | 70.7 |
| Ours | **74.8** | **64.5** | **85.8** | **84.8** | **77.5** |

(b)

| Methods | →Ar | →Cl | →Pr | →Re | Avg. |
|---------|------|------|------|------|------|
| RS | 70.4 | 61.2 | 82.9 | 81.1 | 73.9 |
| MaxP | 70.1 | 61.8 | 83.2 | 81.2 | 74.1 |
| Ours | **74.8** | **64.5** | **85.8** | **84.8** | **77.5** |

## F    COMPLEXITY ANALYSIS

Although the proposed method uses two additional phases with the warm up and pseudo label refinement, the proposed method remains relatively lightweight. We outlined the profiles of computational complexity for each phase in Table 10, by calculating the proportion of each phase to the total training time. The target adaptation phase commonly used with the previous studies consumed the majority of the training time. Specifically, in the Office dataset, the label refinement and the warm-up phase took 8.6 minutes and only 0.2 minutes, while target adaptation required 26.6 minutes. For the DomainNet dataset, the refinement and warm up phase consumed 138 and 5 minutes, respectively. In the other hand, the target adaptation consumed 277 minutes. The shorter training time for PRN can be attributed to the fewer training parameters in PRN compared to the target model (e.g. ResNet101 used in the table). On average, the PRN has only approximately 0.33 % of the parameters of the target model. It uses 0.14M parameters for the DomainNet dataset.

Table 10: Comparison of training time (%) in different phases.

| Benchmarks | Warmup | Label Refinement | Target Adaptation |
|---|---|---|---|
| Office | 0.5% | 24.3% | 75.3% |
| Office-Caltech | 0.4% | 29.0% | 70.6% |
| Office-Home | 0.7% | 28.8% | 70.6% |
| DomainNet | 1.2% | 32.8% | 66.0% |

## G  LIMITATION

We presented a method on how to select $p_c$, and the proposed method achieves better performance than compared methods as described in Section 4.3. Nevertheless, we reveal that the method does not guarantee that the $p_c$ would reach the minimal value of the risk, theoretically. The problem is caused by an inherent error $\epsilon(p_c, y_T)$ in Theorem 2, in which $y_T$ is unknown. However, in fact, this is a common problem in all the BDA scenarios.

## H  CODE DISTRIBUTION

All the codes and instructions for implementation are enclosed with the supplementary material.

