# OpenReview forum: "Label Space-Induced Pseudo Label Refinement for Multi-Source Black-Box Domain Adaptation"
_ICLR.cc/2024/Conference — ICLR 2024 Conference Withdrawn Submission_

### Official Review · Reviewer_mZ6x · 2023-10-30

**Soundness:** 2 fair
**Presentation:** 1 poor
**Contribution:** 2 fair
**Rating:** 3
**Confidence:** 5

**Summary:**

This work studies multi-source black-box domain adaptation where the user adapts multiple source models (parameters unaccessible) to a shifted target domain. In particular, the authors first modify the classical triangle inequality in multiple source domain adaptation by relating the upper bound with the best source model. Then, the authors propose to divide the label space by concentration and closeness. Thereafter, they introduce a pseudo-label refinement network via cross-attention between prediction and the original source labels. Experimental results show that the proposed method obtains better performance than the baselines.

**Strengths:**

1. The studied problem is interesting and significant for distilling black-box models into user-accessible models.

2. The authors provide thorough experimental details for reproducing this paper.

**Weaknesses:**

1. The major concern is that the presentation is **terrible**. It's hard to understand the motivation and the details of this work. In fact, I don't think this can be easily fixed during the rebuttal phase.

2. First, the author mentioned that existing BDA methods suffer from 'decrease empirical errors e.g. through de-noising of source predictions and label smoothing when yT is not accessible'. But, in the modified triangle inequality, there is still a term of p_c, whose definition requires the ground truth.

3. I don't quite understand how p_c can be properly obtained. It seems this paper attempts to obtain p_c via the concentration of one source's predictions to those of other sources. What if there is only one best source while the other sources are all terrible? Also, Section 3.3 is confusing. Why the label space should be divided?

4. The proposed label refine network also confused me. Is the network architecture derived according to the theorem? I don't think such a complex network in the label space can improve the performance.

5. The final prediction performance improvement is marginal. Also, a recent SOTA paper is missing - https://openreview.net/forum?id=hVrXUps3LFA

**Questions:**

See above.

---

### Official Review · Reviewer_xsDH · 2023-10-31

**Soundness:** 2 fair
**Presentation:** 1 poor
**Contribution:** 2 fair
**Rating:** 1
**Confidence:** 5

**Summary:**

This paper proposes a method for black-box source-free domain adaptation where multiple source models exist. The key idea lies in the label refinery of coarse pseudo labels from multiple source models. Results show that the proposed method achieves better performance across a variety of domain adaptation datasets.

**Strengths:**

- The studied problem is relatively new and requires more attention.

- The results on multi-source-free domain adaptation sound better than previous methods.

**Weaknesses:**

- First of all, this paper is poorly written, making it fairly hard to follow. 1) the definitions and explanations are not easy to understand. 2). the idea behind the proposed method is also not easy to understand.

- Concerning the novelty of the proposed method, the core contribution lies in the refinement of coarse pseudo labels from multiple source models, with the target adaptation the same as previous methods.

- In the experiments, the results are not impressive when compared with DINE (CVPR 2022). Several new black-box source-free methods are missing in the paper.

- There exist too many hyper-parameters like \lambda_ in the objective, how to determine them for an unsupervised adaptation problem?

**Questions:**

see the weaknesses above

Sec 3.3 is not easy to follow, the authors are suggested to explain these variables more clearly. For example, a pytorch-like pseudo-code or an illustrative picture is welcome.

I_c denotes the indices of the source model that corresponds to the voting decision, but how about "p_d is chosen as min p_{i_dj'} over i_d"? why does there exist a minimum operation?

Concerning the objective in Eq.(6), how to avoid a non-trivial solution where the PRN degenerates into an identity operation? Also, the results in Table 3(a) are strange, why does the second row under-perform the first row, with the incorporation of the warmup stage?

In Eq.(8), is P_c frozen through the label refinement stage? And the symbols are confusing in Eq.(8-9), is "p^r \in P_c" correct?

---

### Official Review · Reviewer_6ppz · 2023-11-01

**Soundness:** 3 good
**Presentation:** 2 fair
**Contribution:** 2 fair
**Rating:** 3
**Confidence:** 3

**Summary:**

In this paper, the authors proposed a MSBDA pseudo-labeling framework. It incorporates a Pseudo label Refinery Network (PRN) that learns the relation between each source conditioned by the target from source predictions. The target model is adapted by self-learning using a pseudo label generated by PRN. The paper provides theoretical proofs of the core method and validates the effectiveness of the proposed approach through comparative experiments on multiple datasets.

**Strengths:**

（1）The motivation of this paper is quite intriguing. It approaches the problem from a new perspective of label space segmentation and proposes a method to enhance the quality of pseudo-labels, thereby improving the performance of the target model.
（2）This paper provides theoretical proofs related to the proposed method.
（3）This paper not only compares the proposed framework with state-of-the-art methods on multiple datasets but also conducts comprehensive ablation experiments.

**Weaknesses:**

（1）This paper has some writing issues, particularly regarding its overall logical structure and certain details. For example, the paper explains the meanings of some symbols in the main text, while others are not directly defined, and the authors put the symbol definition in the appendix without providing appropriate in-text references；Moreover, there are also problems of symbol abuse in this paper, which set up a lot of unnecessary obstacles for readers in comprehending and interpreting the content. Section 3.3 is where these problems are prominently reflected：The paper uses 'i' to represent the i-th source model, but the definition of the set 'I' is not provided (it's also not included in the appendix). Furthermore, the definition of the set 'Ic' is not standardized. In this context, I understand 'Ic' to represent the set of indexes of source models predicted as label 'c'，the value of 'jc*' should be 'c', and it's possible that some of the symbol definitions are redundant or unnecessary. Additionally, there might be an issue with the definition of 'j''. Furthermore, when the cardinality of 'Ic' is 'M', there is another possible scenario where all source models have identical predictions. The paper does not explicitly mention this, which is a lack of rigor. The definition of 'P' as a set is provided in the appendix, but it lacks standardization.
（2）In Section 4.2 of this paper, where the comparison with other SOTA methods is presented, only results are provided, and there is a lack of analysis.
（3）This paper lacks certain essential details. For instance, it mentions the use of dropout in the model but does not specify the dropout probability. Additionally, it states the use of ResNet-101 as the target model, but it doesn't clarify whether this network was pretrained.

**Questions:**

The paper's motivation is very interesting, and it not only proposes a complete methodological framework but also provides relevant theoretical proofs. However, in contrast, there is significant room for improvement in the writing of the paper. The content lacks a reasonable overall organization of logic and has many issues in detail. The paper's overall completeness is not ideal. Therefore, it is recommended that the authors address these issues by providing clearer definitions and improving the overall structure of the paper to enhance its readability and comprehension.

---

### Official Review · Reviewer_rvLQ · 2023-11-06

**Soundness:** 2 fair
**Presentation:** 2 fair
**Contribution:** 2 fair
**Rating:** 3
**Confidence:** 5

**Summary:**

This paper proposes a novel MSBDA pseudo-labeling framework that used only the predictions of source models to explore positive knowledge from multiple source domains. The performance of the proposed method is evaluated on various benchmark datasets.

**Strengths:**

1. A novel MSBDA framework that leverages only the predictions of source models to explore positive knowledge from multiple source domains is developed.
2.  A theoretical analysis is present to demonstrate the effectiveness of the proposed training strategy.
3. The authors evaluate the proposed method on four benchmark datasets and demonstrate this effectiveness.

**Weaknesses:**

1. The authors claims that the existing MSBDA studies ignore the difference importance of source models. How to illustrate the importance of each source model. By the way, which is the difference with CAIDA when evaluating the importance of each source.
2. In the experiment, the results cannot signifcantly achieve the best results when comparing with the SOTAs, such as CAiDA in the Office-Caltech case.
3.  Regarding the theoretical analysis, how to prove that multi-source could improve the performance of target model?
4.  The current manuscript should be well polished, regarding the tense and presention.

**Questions:**

Please check the Weaknesses above.